# Methoxylated Flavonols and *ent*-Kaurane Diterpenes from the South African *Helichrysum rutilans* and Their Cosmetic Potential

**DOI:** 10.3390/plants12152870

**Published:** 2023-08-04

**Authors:** Olugbenga K. Popoola, Jeanine L. Marnewick, Emmanuel I. Iwuoha, Ahmed A. Hussein

**Affiliations:** 1Applied Microbial and Health Biotechnology Institute, Cape Peninsula University of Technology, Bellville 7535, South Africa; marnewickj@cput.ac.za; 2Chemistry Department, University of the Western Cape, Private Bag X17, Bellville 7535, South Africa; eiwuoha@uwc.ac.za; 3Chemistry Department, Cape Peninsula University of Technology, Symphony Rd., Bellville 7535, South Africa; mohammedam@cput.ac.za

**Keywords:** *Helichrysum rutilans*, phytochemicals, flavonols, diterpenes, skin aging, antioxidant, anti-tyrosinase

## Abstract

Chromatographic fractionation of a methanol extract of *Helichrysum rutilans* afforded seven known compounds. The isolated compounds were identified as 5,7,8-trihydroxy-3,6-dimethoxyflavone-8-*O*-2-methyl-2-butanoate (C-**1**), 5,7-dihydroxy-3,6,8-trimethoxyflavone (C-**2**), 5-hydroxy-3,6,7,8-tetramethoxyflavone (C-**3**), 5-hydroxy-3,6,7-trimethoxyflavone (C-**4**), *ent*-kaurenoic acid (C-**5**), *ent*-kauran-18-al (C-**6**), and 15-α-hydroxy-(-)-*ent*-kaur-16-en-19-oic acid (C-**7**). Compounds C-**1**–C-**4** demonstrated high antioxidant capacities on ORAC hydroxyl radical (2.114 ± 4.01; 2.413 ± 6.20; 1.924 ± 16.40; 1.917 ± 3.91) × 10^6^; ORAC peroxyl radical (3.523 ± 3.22; 2.935 ± 0.13; 2.431 ± 8.63; 2.814 ± 5.20) × 10^3^ µMTE/g; and FRAP (1251.45 ± 4.18; 1402.62 ± 5.77) µMAAE/g, respectively. Moderate inhibitory activities against Fe^2+^-induced lipid peroxidation were observed for C**-1**–C**-4** as IC_50_ values of 13.123 ± 0.34, 16.421 ± 0.92, 11.64 ± 1.72, 14.90 ± 0.06 µg/mL, respectively, while their respective anti-tyrosinase activities with IC_50_ values of 25.735 ± 9.62, 24.062 ± 0.61, 39.03 ± 13.12, 37.67 ± 0.98 µg/mL were also observed. All compounds demonstrated TEAC values within the range of 1105–1424 µMTE/g. The result is an indication that a methanol extract of *H. rutilans* might possibly be a good source of natural antioxidants against ailments caused by cellular oxidative stress and as inhibitors against skin depigmentation, as well as possible raw materials needed for slowing down perishable agricultural products. This is the first report on the phytochemical and biological evaluation of *H. rutilans*.

## 1. Introduction

Oxygen serves as an important molecule for various cellular and metabolic processes. Nevertheless, this beneficial molecule can also be detrimental for living cells at a threshold level of accumulation. Disequilibrium between the cellular production and the oxidative stress clearing process may trigger the production of various free radical molecules [1]. In addition, persistent exposure to ultra-violet (UV) radiation generated primarily from sunlight and other activities (such as the use of sun/tanning beds) may result in premature/untimely skin aging (photoaging), characterized by uneven formation of pigments and loss of structural integrity of the skin, and also in damage to perishable agricultural products [2,3].

Considering various detrimental impacts of free radicals on essential elements in the human body, extensive research on naturally occurring phytochemicals with antioxidant properties is becoming a point of great interest. One such natural phytochemical group of compounds with notable antioxidant activity is flavonoids, with flavones, an important subgroup of flavonoids, predominantly distributed in the plant kingdom [4]. Evidence based on pharmacological data has shown that flavones are well-known for their beneficial health effects and wide applications in the nutraceutical, pharmaceutical, cosmetics/cosmeceutical, and medicinal sectors [5]. These beneficial effects can be structurally and biologically attributed to their antioxidant and anti-inflammatory properties, coupled with their capacities to modulate the activities of key cellular enzymes [6]. Diterpenes, on the other hand, are natural products structurally based on a podocarpane carbon skeleton, with only moderate antioxidant activity, with the exception of some diterpenes having multiple phenolic attachments [7]. A previous review reported on the established ethnomedicinal significance of diterpenes, ranging from antioxidant and anticancer activities and roles as mediators in metabolic diseases [8].

*Helichrysum rutilans* is a dense twiggy shrublet (≤600 mm tall) with characteristic lemon-yellow bracts. It occurs in sandy or stony soils of the Western and Eastern Cape Provinces and southern parts of the Free State in South Africa [9]. No traditional or scientific information about this plant has been documented in the literature according to SciFinder and the dictionary of natural products databases (as of July 2022). Although a great deal of research has been carried out on the genus *Helichrysum*, registered background information on *H. rutilans* is lacking with respect to its scientific and medicinal usage. This research was connected to the motivation for the medicinal possibilities of *H. rutilans* to possess phytochemicals with notable biological activities, due to previous scientific information documented on numerous phytochemicals isolated from South African Helichrysum species with varying degrees of biological and pharmacological importance [10].

The current study aims to elucidate the phytochemical and biological significance of secondary metabolites produced by *H. rutilans*.

## 2. Materials and Methods

### 2.1. Reagents and Organic Solvents

Organic solvents including methanol, acetonitrile (HPLC graded), ethanol, ethyl acetate, dichloromethane, n-hexane (redistilled), vanillin, deuterated chloroform, and acetone were purchased from Merck (Darmstadt, Germany). Sulphuric acid, acetic acid, silica gel 60H (0.040–0.063 mm), sephadex LH-20, pre-coated TLC plate (silica gel, 60F_254_), sodium acetate, hydrochloric acid, ascorbic acid, potassium hydrogen phosphate (K_2_HPO_4_), potassium dihydrogen phosphate (KH_2_PO_4_), potassium chloride, iron II tetraoxosulphate VI, tetrachloro acetic acid (TCA), butylated hydroxytoluene (BHT), ethylenediammine tetraacetic acid (EDTA), thiobarbituric acid (TBA), dimethyl sulphoxide (DMSO), tris (hydroxymethyl) aminoethane hydrochloride (tris HCl) were secured from Kimix (Cape Town, South Africa). Standards (purity > 99.0%) for antioxidant activities, inhibition of Fe^2+^-induced lipid peroxidation, anti-elastase and anti-tyrosinase assays such as kojic acid, oleanolic acid, epigallocatechin gallate (EGCG), 6-hydroxyl-2,5,7,8-tetramethylchroman-2-carboxylic acid (trolox), 2,2′-Azino-bis (3-ethylbenzo thiazoline-6-sulfonic acid) diammonium salt (ABTS), potassium peroxodisulphate (K_2_S_2_O_8_), fluorescein sodium salt, 2,2′-Azobis (2-methylpropionamidine) dihydrochloride (AAPH), perchloric acid, 2,4,6-tri[2-pyridyl]-s-triazine (TPTZ), Iron (III) chloride hexahydrate (FeCl_3_·6H_2_O), sepharose (wet bed diameter, 60–200 µm), copper (II) tetraoxosulphate VI (CUSO_4_), hydrogen peroxide, N-succ-(Ala)3-nitroanilide (SANA), mushroom tyrosinase, elastatse (from porcine pancrease), and L-tyrosine were all secured from Sigma-Aldrich, Inc. (St. Louis, MO, USA). All antioxidant assays, including FRAP, TEAC, lipid peroxidation, and skin enzyme inhibition (tyrosinase and elastase), were measured using a Multiskan Spectrum Plate Reader (Thermo Fisher Scientific, Waltham, MA, USA), while automated ORAC assays were read using a Floroskan Spectrum Plate Reader (Thermo Fisher Scientific, Waltham, MA, USA). All isolation and purification was carried out using high-performance liquid chromatography HPLC (Agilent Technologies 1200 series, Santa Clara, CA, USA) equipped with UV detector, manual injector, quaternary pump (G1311A), vacuum degasser (G1322A), column compartment (G1316A), and reversed-phase C18 column (SUPELCO, 25 × 2.1 cm, 5 µm, Sigma-Aldrich, St. Louis, MO, USA).

### 2.2. Preparation of Plant Extracts

The plant material was collected in October 2012 from Jonkershoek (approx. 9 km SE Stellenbosch) Nature reserves, Western Cape, South Africa. The voucher specimen was identified by Prof. Christopher Cupido (SANBI, Kirstenbosch), and a copy has been deposited at the Compton Herbarium, South African National Biodiversity Institute, Kirstenbosch, South Africa, with herbarium number NBG145883.

### 2.3. Extraction and Purification of Chemical Constituents

The aerial part of the plant material (400 g) was air-dried at room temperature, blended, and extracted with methanol at room temperature (25 °C) for 48 h. The methanol extract was evaporated until dryness using a rotary evaporator at 40 °C to yield 19.0 g (4.75%) of residue. The total extract (15.5 g) was applied to a silica gel column (30 × 18 cm) and eluted using a gradient of hexane (Hx) and ethyl acetate (EtOAc) in the following order of increasing polarity: 100% Hx, Hx-EtOAc (9:1, *v*/*v*), (85:15, *v*/*v*), (4:1, *v*/*v*), (7:3, *v*/*v*), (3:2, *v*/*v*), and (1:1, *v*/*v*). The collected fractions (1–63, 250 mL each, Figure 1a) were concentrated on the rotary evaporator and combined according to their TLC (using solvent systems A–D, Figure 1a) profiles. The column fractions were combined to form sixteen (I–XVI, Figure 1b) representative main fractions with gradient of solvent systems (ss): hexane:ethylacetate (*v*/*v*) as follows: I (10–14, ss 9:1); II (15, 16, ss 9:1); III (17–20, ss 85:15); IV (21–23, ss 85:15); V (24–26, ss 4:1); VI (27, 28, ss 4:1); VII (29–32, ss 4:1); VIII (33–36, ss 7:3); IX (37, ss 7:3); X (38, 39, ss 7:3); XI (40–43, ss 7:3); XII (44–48, ss 3:2); XIII (49–53, ss 3:2); XIV (54–56, ss 1:1); XV (57–60, ss 1:1); XVI (61, ss 1:1).

The crystal of main fraction X (411 mg), when washed with dichloromethane (DCM) followed by methanol and finally developed on TLC using solvent system E (XA_ppt_. in Figure 1c), afforded C**-7**, a white crystal identified as 15-α-hydroxy-(-)-kaur-16-en-19-oic acid (C-**7**, 90 mg, 0.0225%). Methanolic supernatant XC (70 mg) obtained from main fraction X was rechromatographed on sephadex with isocratic elution with 20% aq. ethanol (*v*/*v*) and developed on TLC (using solvent system F, Figure 1d) to give sub fractions XC1 (29 mg). XC1 (25 mg) was injected into the HPLC and eluted using gradient solvent system of ACN and de-ionized water (55:45 to 65% ACN in 30 min, then 100% in 15 min). One prominent peak collected as displayed in Figure 1e was identified as 5,7-dihydroxy-3,6,8-trimethoxyflavone (C-**2**, Rt 26 min, 18 mg, 0.0045%). Main fraction VIII (291 mg) was re-chromatographed on sephadex using isocratic elution of 10% aq. ethanol; the collected fractions (50 mL each) were combined in accordance with their TLC (ssF, Figure 1f) profiles to yield sub fraction VIIIA. VIIIA (90 mg) was injected into the HPLC (50:50 to 70% ACN in 30 min, then 100% in 15 min, Figure 1g). One prominent peak collected was identified as 5,7,8-trihydroxy-3,6-dimethoxyflavone-8-*O*-2-methyl-2-butanoate (C-**1**, Rt 37 min, 22 mg, 0.0055%). Fraction VI (192 mg) was re-chromatographed on silica using isocratic elution of 1% methanol in DCM; the collected fractions were combined according to their TLC (ssE, Figure Ih) profiles to yield subfraction VIB, which was injected into the HPLC (70:30 to 80% ACN in 30 min, then 100% in 15 min, Figure 1i). Two prominent peaks collected were identified as 5-hydroxy-3,6,7-trimethoxyflavone (C-**4**, Rt 18 min, 12 mg, 0.003%) and 5-hydroxy-3,6,7,8-tetramethoxyflavone (C-**3**, Rt 21 min, 22 mg, 0.0055%). Crystals formed from main fractions II and III were separately washed with hexane. White needle-like crystals were obtained and identified as ent-kaurenoic acid (C-**5**, 62 mg, 0.0155%) and ent-kauran-18-al (**C**-**6**, 65 mg, 0.01625%), respectively.

### 2.4. Antioxidant and Biological Characterization

#### 2.4.1. Ferric-Ion-Reducing Antioxidant Power (FRAP) Assay

Working FRAP reagent was prepared in accordance with the method described previously [11]. In a 96-well clear microplate (visible range), 10 μL of the stock solution (1 mg/mL, *w*/*v*) of the isolated compounds (C-**1**–C-**7**) and a methanol extract (HR) were mixed each with 300 μL FRAP reagent. The FRAP reagent was a mixture (10:1:1, *v*/*v*/*v*) of acetate buffer (300 mM, pH 3.6), tripyridyl triazine (TPTZ) (10 mM in 40 mM HCl), and FeCl_3_·6H_2_O (20 mM). After incubation at room temperature for 30 min, the plate was read at a wavelength of 593 nm on a Multiskan spectrum plate reader. L-Ascorbic acid was used as a standard with concentrations varying between 0 and 1000 μM. Further dilutions were carried out on the samples that were highly concentrated and such dilution factors were recorded and used for the calculations of the affected samples. The results were expressed as μM ascorbic acid equivalents per milligram dry weight (μM AAE/g) of the test samples.

#### 2.4.2. Automated Oxygen Radical Absorbance Capacity (ORAC) Assay

ORAC was carried out according to the previous method [12] with some modifications [13,14]. The method measures the antioxidant scavenging capacity of thermal decomposition generated by either (i) peroxyl radical of 2,2′-azobis (2-amino-propane) dihydrochloride (AAPH; ORAC_ROO**·**_ assay) or (ii) hydroxyl radical (ORAC_OH**·**_ assay), generated by H_2_O_2_-Cu^2+^ (H_2_O_2_, 0.3% *m*/*v*; Cu^2+^ [as CuSO_4_], 18 µM, at 37 °C. Fluorescein was used as the fluorescent probe. The loss of fluorescence of fluorescein was an indication of the extent of its oxidation through reaction with the peroxyl or the hydroxyl radical. The protective effect of an antioxidant is measured by comparing the fluorescence area under the curve (AUC) plot relative to that of a blank in which no antioxidant was present. The analyzer was programmed to record the fluorescence of fluorescein every 2 min after AAPH, or H_2_O_2_-Cu^2+^ was added. The fluorescein solution and sample were added to the wells of an illuminated 96-well plate, and 12 µL of each of our samples (in stock solution of 1 mg/mL) was combined with 138 µL of a fluorescein working solution followed by the addition of 50 µL of 150 mg of AAPH prepared in situ in 6 mL phosphate buffer. Absorbance was measured using the Fluoroskan spectrum plate reader with the excitation wavelength set at 485 nm and the emission wavelength at 530 nm. A calibration curve was established using a trolox stock solution of concentrations in the range of 83–417 µM (R^2^ = 0.9514). The ORAC values were calculated using a regression equation (Y = a + bX + cX^2^) between trolox concentration (Y in µM) and the net area under the fluorescence decay curve (X). ORAC values were expressed as micromoles of trolox equivalents (TEs) per milligram of test sample. Samples without a perfect curve were further diluted and the dilution factors were used for the result calculations of such samples.

#### 2.4.3. Trolox Equivalent Absorbance Capacity (TEAC) Assay

The total antioxidant activity of the test sample was measured using previously described methods [15,16]. The stock solutions which contained 7 mM ABTS and 140 mM potassium-peroxodisulphate (K_2_S_2_O_8_) (Merck, Johannesburg, South Africa) were prepared and kept at −2 °C. The working solution was then prepared by adding 88 μL K_2_S_2_O_8_ solution to 5 mL ABTS solution. The two solutions were thoroughly mixed and allowed to react for 24 h at room temperature in the dark. Trolox (6-hydrox-2,5,7,8-tetramethylchroman-2-carboxylic acid) was used as the standard with concentrations ranging between 0 and 500 μM. After 24 h, the ABTS mix solution was diluted with ethanol to read a start-up absorbance (control) of approximately 2.0 (±0.1). The stock solution (1 mg/mL) of a methanol extract (HT) and purified compounds (25 μL) were allowed to react with 300 μL ABTS in the dark at room temperature for 30 min. The absorbance was read at 734 nm at 25 °C on the plate reader. The results were expressed as μM trolox equivalents per milligram dry weight (μM TE/g) of the test samples.

#### 2.4.4. Inhibition of Fe (II)-Induced Microsomal Lipid Peroxidation Assay

A previously described method [17] was used with a few modifications. The reaction mixture contained microsomes (1 mg of protein/mL in 0.01 M potassium phosphate buffer; pH 7.4, supplemented with 1.15% KCl, *m*/*v*) isolated from an S9 liver fraction using a sepharose column. The positive control included microsomes, buffer, and ferrous sulphate, in the absence of the samples to be tested.

The sample stock solutions (HR and C-**1**–C-**7**) were prepared in methanol (1 mg/mL, *w*/*v*). The working sample solutions were prepared in 0.01M potassium phosphate buffer, pH 7.4, supplemented with 1.15% KCl (*m*/*v*) diluted to 26.750, 13.375, 6.688, 3.344, 1.672, and 0.836 µg/mL concentrations. A total of 100 µL of each sample (working solutions) was dissolved in potassium phosphate buffer and pre-incubated with 500 µL microsomes at 37 °C for 30 min in a shaking water bath. A total of 200 µL of KCl buffer was added to the mixture, followed by 200 µL of a 2.5 mM ferrous sulphate solution and incubated at 37 °C for 1 h in a shaking water bath. The reaction was terminated with 10% trichloroacetic acid (TCA, *m*/*v*) solution (1 mL) containing 125 µL butylated hydroxytoluene (BHT, 0.01%) and 1 mM ethylenediaminetetraacetic acid (EDTA). Samples were centrifuged at 2000 rpm for 15 min; 1 mL of supernatant was mixed with 1 mL of 0.67% thiobarbituric acid (TBA) solution. The reaction mixture was then incubated in a water bath at 90 °C for 20 min and the absorbance was measured at 532 nm using Multiskan plate reader. EGCG, dissolved in methanol, was used as positive control. The percentage inhibition of TBARS formation relative to the positive control was calculated by
$$\frac{A_{control}-A_{sample}}{A_{control}}\times 100$$

#### 2.4.5. Tyrosinase Enzyme Assay

This assay was performed using previously described methods [18,19] with slight modifications. Samples were dissolved in DMSO (dimethyl sulphoxide) to a stock solution of 1 mg/mL (*w*/*v*). Further dilutions were carried out with 50 mM sodium phosphate buffer (pH 6.5) for all working solutions to the concentrations of 100.00, 50.00, 25.00, 12.50, 6.25, 3.12, and 1.56 µg/mL. Kojic acid dissolved in DMSO was used as a control drug. In the wells of a 96-well plate, 70 µL of each sample of working solution was combined with 30 µL of tyrosinase (Sigma-Aldrich CAS No. 9002-10-2, from mushroom, 500 Units/mL in sodium phosphate buffer) in triplicate. After incubation at room temperature for 5 min, 110 µL of substrate (2 mM L-tyrosine, Sigma-Aldrich, CAS No. 35424-81-8) dissolved in sodium phosphate buffer was added to each well. Final concentrations of the crude extract, isolated compounds, and positive control (kojic acid) ranged from 1.0 to 100 µg/mL (*m*/*v*). The sample control was made up of each sample with sodium phosphate buffer. The reacting mixture was then incubated for 30 min at room temperature. The enzyme activity was determined by measuring the absorbance at 490 nm using a plate reader. The percentage of tyrosinase inhibition was calculated as follows:$$\frac{\left(A-B\right)-\left(C-D\right)}{\left(A-B\right)}\times 100$$
where *A* is the absorbance of the control with the enzyme, *B* is the absorbance of the control without the enzyme, *C* is the absorbance of the test sample with the enzyme, and *D* is the absorbance of the test sample without the enzyme.

#### 2.4.6. Elastase Inhibition Assay

Inhibition of elastase by the test samples was assayed using N-succ-(Ala)3-nitroanilide (SANA) as the substrate, monitoring the release of *p*-nitroanilide by the method described [18] with little adjustment. The inhibitory activity determined the intensity of color released during cleavage of SANA by the action of elastase. The working reagent for anti-elastase assay was prepared in line with the literature. The preparation involved 1 mM SANA (Sigma-Aldrich, CAS No. 108322-03-8) in 0.1 M tris-HCl buffer pH 8.0. The sample stock solutions were prepared as 1 mg in 1 mL methanol. A total of 200 µL of already prepared SANA was added to the 20 µL of sample solution (diluted to 100.00, 50.00, 25.00, 12.50, 6.25, and 3.12 µg/mL concentrations in tris-HCl buffer as working solutions) in a 96-well plate. The mixtures were vortexed and pre-incubated for 10 min at 25 °C and then 40 µL of elastase from porcine pancrease (Sigma-Aldrich, CAS No. 39445-21-1, 0.03 Units/mL) prepared in Tris-HCl buffer was added. The mixtures were further incubated for 10 min and the absorbance was measured at 410 nm. Tris-HCl buffer was used as control, while oleanolic acid dissolved in methanol (1 mg/mL, further diluted to 100.00, 50.00, 25.00, 12.50, 6.25, and 3.12 µg/mL concentrations in tris-HCl buffer) was used as a positive control. The sample control involved the sample and SANA (uninhibited sample). The percentage of elastase inhibition was calculated as follows:$$\frac{1-B}{A}\times 100$$

Elastase inhibition (%), where *A* is the enzyme activity without sample and *B* is the activity in the presence of the sample.

### 2.5. Statistical Analysis

All skin enzyme inhibitory assays and Fe^2+^-induced lipid peroxidation assay calculations, expressed as percentage inhibitions ± SD, were performed using MS Excel 2013, while the final values, expressed as IC_50_, were determined using GraphPad prism 5.0. The data presented are means ± SD obtained from 96-well plate readers for all in vitro experiments in triplicate. Differences between the means were considered to be significant if *p* < 0.05 according to Prism’s one-way ANOVA. The ORAC, FRAP, and TEAC values were determined using their respective templates. Correlation coefficients among the methods used for determining total antioxidant assays (FRAP, TEAC, and ORAC), as well as skin enzymes inhibitions (tyrosinase anf elastase), were determined using SPSS version 21. Values at *p* < 0.05 were considered significant.

## 3. Results

### 3.1. Spectroscopic Measurements

Chromatographic purification of methanol extract of *H. rutilans* (HR) using different chromatographic techniques, including semi-prep HPLC, resulted in the isolation of seven (7) compounds in a pure state, categorized into methoxylated flavonols (C-**1**–C-**4**) and kaurane diterpene derivatives (C-**5**–C-**7**). Compound C-**1** was isolated as a yellow solid (UV λ_max_ (MeOH) nm: 280, 360; IR (KBr) cm^−1^: 3300, 1800, 1600, and 1150; HRMS *m*/*z* 413.1201 [M+1]^+^ (calcd. 413.1192)). HRMS established the molecular formula of C-**1** as C_22_H_20_O_8_ (*m*/*z* 413.1201). UV spectrum showed absorption at λ_max_ 280 nm and 360 nm (MeOH), in addition to dark purple spots under short UV (254 nm) as an indication of flavonoid characteristics. IR (KBr) spectrum showed bands of a hydroxyl group (3300 cm^−1^), carbonyl (1800 cm^−1^), conjugated C=C 1600 cm^−1^), and C-O side chain substituent (1150 cm^−1^). ^1^H- and ^13^C-NMR measurements of C-**1**–C-**4** are indicated in Table 1.

### 3.2. Total Antioxidant Capacities

The total antioxidant capacities of both the methanol extract of *Helichrysum rutilans* (HR), representing the total amount of extracted substances present in the plant material, and its isolated compounds (C-**1**–C-**7**) were investigated in an in vitro system, using varying degrees of spectrophotometric measurements (FRAP, TEAC, ORAC, LPO). All investigated samples were in accordance with SOPs and the analyzed results are indicated in Table 2.

### 3.3. Skin Enzyme Inhibitory Activities

End point measurements of skin enzyme inhibitory activities (tyrosinase and elastase) were investigated after an incubating period of tyrosinase with test samples in the presence of substrate at different sample concentrations. The results obtained (Table 3) after absorbance measurement at 490 nm were analyzed using GraphPad prism V5.0 of triplicate values.

## 4. Discussion

NMR spectra (Table 1) showed 22 carbon signals, 15 of them belonging to flavonoid skeleton with unsubstituted ring B [δ_H_ 8.02, *br s*, 2H; 7.29, *br s*, 2H; and 7.29, *br s*, 1H; δ_C_: 128.1 (2XC), 128.4 (X 2C), and 130.9 (1C)], with signals of two methoxyls at δ_H_ 3.88 and 4.11 and a 5-OH signal at 12.65. There were additional signals of a tigloyl side chain esterified with a hydroxyl group of ring A, two olefinic methyls at δ_H_ 2.13 (*d*, 6.8 Hz; δ_C_ 15.8), 2.16 (*s*; δ_C_ 20.3), and an olefinic proton at 6.39 (*q*, 6.8 Hz; δ_C_ 140.9). The above data indicated the presence of a flavonol skeleton with two methoxyl and a tigloyl ester groups. The positions of the two methoxyls were evidenced from HMBC correlations which showed cross-peaks between *O*-methyl (δ_H_ 3.88) with C-**3** (139.1) and *O*-methyl (δ_H_ 4.11) with C-**6** (131.2). The fact that both C-**5** and C-**7** contained a free hydroxyl group is evidenced by the OH signal at 12.65 (5-OH) and the C-**7** chemical shift at 149.6. The extra side chain of the tigloyl group was positioned at C8 from the high field shift of C8 to 118.2 ppm. The E geometry of the tigloyl group was supported by the coupling of the H-3″ with methyl 5″ (*J* = 6.8 Hz). The above data established the structure of C**-1** as 5,7,8-trihydroxy-3,6-dimethoxyflavone-8-*O*-2-methyl-2-butanoate (Figure 2), which was confirmed by comparison of the NMR data with the reported values [20]. Isolation of C**-1** was reported once from *Pseudognaphalium cheiranthyfolium* collected in Chile.

Compound C**-2** was identified as 5,7-dihydroxy-3,6,8-trimethoxyflavone [19,20,21] and was isolated previously from *H. decumbens* [21]. Compound **C-3** was identified as 5-hydroxy-3,6,7,8-tetramethoxyflavone [19,20,21]. This compound was reported previously from *H. cepaloideum* [22]. Compound C-**4** was identified as 5-hydroxy-3,6,7-trimethoxyflavone and reported from *H. decumbens* [21].

Compounds C-**5**–C-**7** showed a typical diterpene’s NMR pattern. Compound C-**5** was identified as *ent*-kaurenoic acid after comparing the [α]_D_ value (−0.078) and the ^13^C NMR data with the literature [23]. C**-5** is a common diterpene and was reported from many *Helichrysum* species e.g., *H. fulvum* [24]. Compound C-**6** was identified as *ent-*kauren-18-al, with occurrence reported previously from *H. pilosellum* [23]. Compound C-**7** was identified as 15-β-hydroxy-(-)-kaur-16-en-19-oic acid, isolated previously from *Mikania vitifolia* [25].

Medicinal plant extracts have the potential for being incorporated in cosmetic formulations due to the presence of secondary metabolites (phytochemicals) that display broad applications as antioxidants, capable of combating free radicals generated within the body [26].

During the last decade, natural antioxidants, particularly phenolic compounds, have been under very close scrutiny as potential therapeutic agents against a wide range of ailments including cardiovascular dysfunctions and aging [27]. The type of flavonoids, the degree of methoxylation, and the number of hydroxyl groups are some of the parameters that determine their antioxidant potentials. In general, the differences in antioxidant activity between polyhydroxylated and polymethoxylated flavonoids are most likely due to differences in both hydrophobicity and molecular planarity [28]. Flavonoid radical stability is thought to be increased by the creation of a completely conjugated electron system. This can be accomplished through structural planarity of the flavonoid due to the presence of a hydroxyl group at the C3 position on the C-ring, resulting in a flavonol backbone structure. Replacement of 3-OH by a methoxy substituent at this position perturbs this conjugation, thus rendering the flavonoids less active as antioxidants than their corresponding OH at C3 [29]. In agreement with literature data, the present study corroborates the view that structural features of flavonols (3-OH) are an important moiety for antioxidant efficacy [30,31,32,33].

The total antioxidant activities (Table 2) confirm C-**1**–C-**4** as natural antioxidants but as less active when compared to their hydroxylated analogues previously reported [32]. Our results obtained on the total antioxidant assays, when compared with their corresponding hydroxylated derivatives, therefore described that the methoxylation of flavonoids obviously weakened the antioxidant activity, as previously observed [33]. Our results are in agreement with the existing data, which demonstrated that the replacement of the C3 hydroxyl by a methoxy group resulted in a reduction in antioxidant activity.

The ability of phenolic compounds to inhibit oxidative damage in lipids was assessed using thiobarbituric acid as a model system. Peroxidation was initiated by the addition of a FeSO_4_-EDTA mixture. It is well-known that a transition metal like iron may generate highly reactive hydroxyl or alkoxyl radicals (Fenton reaction). The result indicated compounds C-**1**–C-**4** to have mild inhibitory activities against the Fe^2+^-induced lipid peroxidation as indicated by the IC_50_ values of 13.123 ± 0.34, 16.421 ± 0.92, 11.64 ± 1.72, 14.90 ± 0.06 µg/mL, respectively. This suggests that C-**1**–C-**4** possess a feasible mechanism for iron-chelating and iron-stabilizing capacity due to the presence of 3-OMe and 5-OH in conjugation with the 4-keto group. It is also an indication that the lone pair of electrons on OMe contributes to the antioxidant activities of compounds investigated. Previous work illustrated the significant role of these features in the more potent antioxidant capacity due to the formation of hydrogen bonding between 5-OH and 4-keto groups, which helps to further stabilize the flavone radicals formed [32,34,35,36,37,38]. Compounds C-**1**–C-**4**, in addition to lipophilicity, may therefore serve as a source of good antioxidants due to their ability to stabilize Fe^2+^, thereby reducing the production of the reactive hydroxyl radical (OH·) during the Fenton reaction [39].

Due to the everyday growing market of cosmeceuticals and nutraceuticals, tyrosinase inhibitors have received special attention because of their ability to alleviate hyperpigmentation and undesirable browning of food products. Efforts are geared towards the sourcing of more suitable options obtained from compounds of natural origin due to their excellent pharmacological properties and commendably high safety margins, demonstrated by tyrosinase inhibitors isolated from nature [40,41]. When tyrosinase enzyme activity is inhibited, melanin production is reduced, resulting in fairer skin [42,43]. Amongst the *H. rutilans* constituents investigated for anti-tyrosinase activity in Table 3, compounds C-**1**–C-**4** demonstrated moderate inhibition of tyrosinase (with respective IC_50_ values of 25.735 ± 9.62, 24.062 ± 0.61, 39.03 ± 13.12, 37.67 ± 0.98 µg/mL), but remained less potent than the control included, kojic acid (IC_50_ = 3.511± 1.44 µg/mL). The presence of conjugation in the C-**1**–C-**4** rings further gives rise to a resonance effect in both rings, thereby providing stability to the respective flavone radicals formed [35,44,45,46]. No significant inhibitory activity was recorded when the samples were tested for their inhibition of elastase activity at the threshold concentration of 100 µg/mL.

## 5. Conclusions

As skin biochemistry and skin aging are very complex processes, it is not surprising that ongoing research efforts are aimed at uncovering more natural phytochemicals that can slow down the detrimental effects of noxious substances, including free radicals produced in the skin. Seven known compounds were isolated and characterized by our previous analytical tools. Flavones have been shown to possess good antioxidant activity and have been implicated as inhibitors of lipid peroxidation. The evidence presented from this study suggests that a dietary intake of flavonoid-containing foods may be of benefit in lowering the risk of certain pathophysiologies associated with free radical-mediated events. This in vitro study showed that a methanol extract of *H. rutilans* can be considered a good antioxidant and skin depigmentation option for the relevant industries. Possible cosmeceutical products aimed against photo-oxidation of the skin can be formulated by including the constituents of *H. rutilans*, but further investigations are first required to ascertain possible undesirable side-effects on the skin. Furthermore, in vivo studies might be required to establish the usage of *H. rutilans* and its isolates in the formulation of cosmetic products. Results from the current study present the first report to be documented on *H. rutilans*, a widespread plant in the coastal parts of the western and mountainous parts of the eastern provinces in South Africa.

## Figures and Tables

**Figure 1 plants-12-02870-f001:**
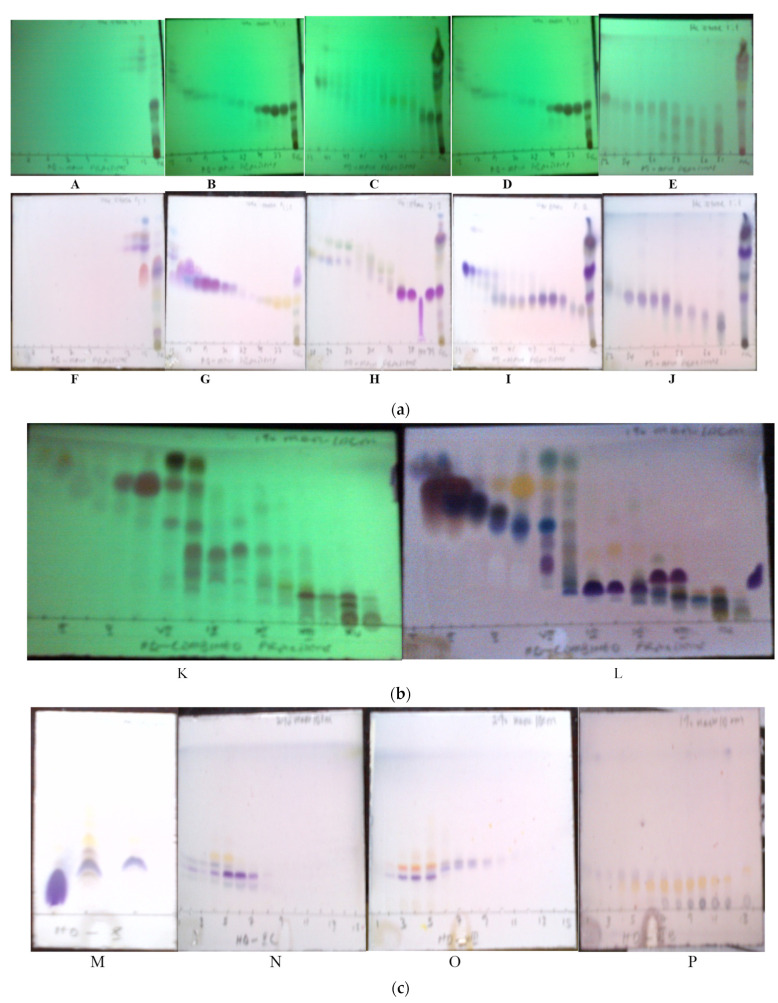
(**a**) TLC profile of the column chromatographic fractions (1–58) under UV (254 nm; **A**–**E**) and after spraying with H_2_SO_4_/vanillin and then being heated (**F**–**J**). TLC plate (**A**,**F**) consisted of fractions 1–15 (ref. to total extract FR) developed using ss A. TLC plate (**B**,**G**) consisted of fractions 15–28 (ref. to total extract FR) developed using ss A. TLC plate (**C**,**H**) consisted of fractions 28–39 (ref. to total extract FR) developed using solvent system B. TLC plate (**D**,**I**) consisted of fractions 39–52 (ref. to total extract FR) developed using solvent system C. TLC plate (**E**,**J**) consisted of fractions 52–61 (ref. to total extract FR) developed using solvent system D. FR refers to methanol total extract of *H. rutilans*. (**b**) TLC profile of the main fractions (I–XVI) under UV (254 nm; (**K**) and after spraying with H_2_SO_4_/vanillin and gentle heating (**L**). (**c**) TLC of X when sprayed with H_2_SO_4_/vanillin and then heated (**M**). TLC profile of XC after spraying with H_2_SO_4_/vanillin and heating (**N**). TLC profile of VIII after spraying with H_2_SO_4_/vanillin and heating (**O**). TLC profile of VI after spraying with H_2_SO_4_/vanillin and heating (**P**). (**d**) HPLC spectrum of C-**2**. (**e**) HPLC spectrum of C-**1**. (**f**) HPLC spectrum of C-**3** and C-**4**.

**Figure 2 plants-12-02870-f002:**
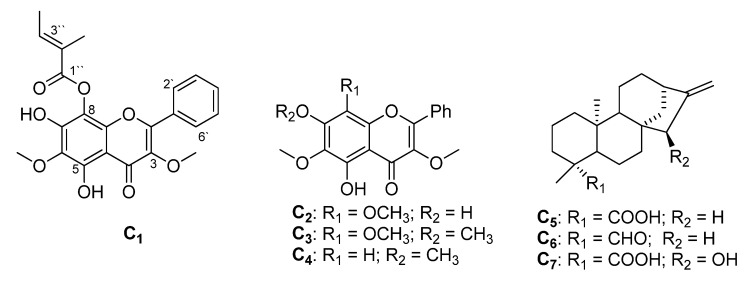
Chemical structures of compounds C-**1**–C-**7** isolated from *H. rutilans*.

**Table 1 plants-12-02870-t001:** ^1^H (400 MHz: m, *J* Hz) and ^13^C (100 MHz) NMR spectral data of isolated compounds **C_1_**–**C_4_** in CDCl_3_.

No.	C-1		C-2		C-3		C-4	
	^13^C	^1^H	^13^C	^1^H	^13^C	^1^H	^13^C	^1^H
2	155.8 *s*		155.8 *s*		156.1 *s*			
3	139.2 *s*		139.3 *s*		139.5 *s*			
4	179.1 *s*		179.5 *s*		179.5 *s*			
5	149.9 *s*		148.9 *s*		153.1 *s*			
6	131.2 *s*		130.5 *s*		130.6 *s*			
7	149.6 *s*		148.1 *s*		149.2 *s*			
8	118.2 *s*		127.2 *s*		136.2 *s*			6.61 *s*
9	144.3 *s*		145.1 *s*		145.1 *s*			
10	104.3 *s*		105.2 *s*		107.6 *s*			
1′	130.1 *s*		128.5 *s*		132.9 *s*			
2′	128.1 *d*	7.29 *br s*	128.3 *d*	8.15 *br s*	128.4 *d*	8.17 *br s*		8.19 *d*, 7.2
3′	128.4 *d*	8.02 *br s*	128.8 *d*	7.65 *br s*	128.8 *d*	7.56 *br s*		7.45 *m*
4′	130.9 *d*	7.29 *br s*	131.1 *d*	7.65 *br s*	131.2 *d*	7.56 *br s*		7.45 *m*
5′	128.4 *d*	8.02 *br s*	128.8 *d*	7.65 *br s*	128.8 *d*	7.56 *br s*		7.45 *m*
6′	128.1 *d*	7.29 *br s*	128.3 *d*	8.15 *br s*	128.4 *d*	8.17 *br s*		8.19 *d*, 7.2
1″	165.5 *s*							
2″	126.4 *s*							
3″	140.9 *d*	6.36 *q*, 6.8						
4″	15.8 *q*	2.13 *d*, 6.8						
5″	20.3 *q*	2.16 *s*						
OMe-3	60.2 *q*	3.88 *s*	60.4 *q*	3.90 *s*	60.4 *q*	3.91 *s*		3.89 *s*
OMe-6	60.6 *q*	4.11 *s*	61.1 *q*	4.09 *s*	61.2 *q*	3.98 *s*		3.92 *s*
OMe-7					62.2 *q*	4.14 *q*		4.06 *s*
OMe-8			61.8 *q*	4.02 *s*	61.7 *q*	3.98 *s*		
5-OH		12.65 *s*		12.55 *s*		12.37		11.37
7-OH		6.58 *br s*		6.80 *br s*				

**Table 2 plants-12-02870-t002:** Total antioxidant capacities of the isolated compounds and methanol extract.

Sample	FRAP µM AAE/g	TEAC µM TE/g	ORAC Peroxyl µM TE/g	ORAC Hydroxyl × 10^6^ µM TE/g	LPO
HR	906.71 ± 5.18	765.23 ± 2.43	2935.16 ± 3.92	1.817 ± 1.72	61.09 ± 4.19
C-**1**	1251.45 ± 4.18	1131.80 ± 6.41	3523.51 ± 3.22	2.114 ± 4.01	13.123 ± 0.34
C-**2**	1402.62 ± 5.77	1276.11 ± 1.32	2935.47 ± 0.13	2.413 ± 6.20	16.42 ± 0.92
C-**3**	1314.42 ± 2.42	1378.10 ± 9.06	2431.30 ± 8.63	1.924 ± 16.40	11.64 ± 1.72
C-**4**	1119.44 ± 11.89	1207.11 ± 7.21	2814.51 ± 5.20	1.917 ± 3.91	14.90 ± 0.06
C-**5**	19.66 ± 8.12	1105.00 ± 3.09	364.44 ± 6.71	0.429 ± 12.00	>100
C-**6**	29.39 ± 5.84	1361.90 ± 0.35	914.29 ± 2.74	0.531 ± 10.24	>100
C-**7**	60.90 ± 7.90	1424.51 ± 0.70	93.10 ± 13.68	0.845 ± 13.34	>100
EGCG	3326.45 ± 5.76	11545.44 ± 17.28	14693.09 ± 5.53	3.862 ± 4.65	0.929 ± 4.11

***p* < 0.05**; **HR**: methanol extract from *H. rutilans*; C-**1**–C-**7**: isolated compounds; **NA**: not applicable; EGCG: Epigallocatechin gallate.

**Table 3 plants-12-02870-t003:** Skin enzyme inhibitory activities displayed by the isolated compounds and methanol extract.

Sample	TYR	ELA
HR	74.15 ± 3.92	>100
C-**1**	25.735 ± 9.62	>100
C-**2**	24.062 ± 0.61	>100
C-**3**	39.03 ± 13.12	>100
C-**4**	37.67 ± 0.98	>100
C-**5**	>100	>100
C-**6**	>100	>100
C-**7**	>100	>100
KJA	3.425 ± 2.37	>100
OLEAN	NA	12.77 ± 0.97

***p* < 0.1. HR**: methanol extract from *H. rutilans*; **C**-**1**–**C**-**7**: isolated compounds; **NA**: not applicable; **KJA**: kojic acid; **TYR**: tyrosinase; **ELA**: elastase; **OLEAN**: oleanolic acid.

## Data Availability

All data are contained within the article. Additional details (nmr spectral) are available from the corresponding author, A.A.H., upon reasonable request.

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
