# Peer review of "Methoxylated Flavonols and *ent*-Kaurane Diterpenes from the South African *Helichrysum rutilans* and Their Cosmetic Potential"

_plants, 2023, doi:10.3390/plants12152870_

Round 1

Reviewer 1 Report

Commnets,

1. Authors should do correlation coefficients among the method used for antioxidant activity (Ferric-Ion Reducing Antioxidant Power (FRAP) Assay, Automated Oxygen Radical Absorbance Capacity (ORAC) Assay, Trolox Equivalent Absorbance Capacity (TEAC) Assay,  and Inhibition of Fe (II)-Induced Microsomal Lipid Peroxidation Assay

2. Also, the same for Tyrosinase Enzyme Assay and Elastase Inhibition Assay.

3. For what (lipid peroxidation) refers to?

4. I didn't see any methods for the chemical characterization of the extract.

All characterizations should be mentioned in the manuscript with the original chromatograms.

Minor errors

Author Response

AUTHORS REPLY TO THE REVIEW REPORT (REVIEWER 1)

Article No. Plants-2491004

Article type: Research

No.

Part

Comments

Author Response

1

Results

Authors should do correlation coefficients among the method used by antioxidant activity FRAP, ORAC, TEAC assay, and inhibition of Fe II-induced lipid peroxidation

1.   Correlation coefficients among the antioxidant assays are significant at p< 0.05 (2 tail measurement) using SPSS version 21. This has been included as footnote on Table 2.

2.   Determination of correlation coefficient has been included in statistical analysis section and highlighted in red

2

Results

Also, the same for tyrosinase and elastase inhibition assay

1.   Table 3 do not correlate at 95% level because most of the test samples were not active with Elastase enzyme. Therefore p< 0.01 (2 tail measurement) using SPSS version 21. This has also been included as footnote on Table 3

2.   Determination of correlation coefficient has been included in statistical analysis section and highlighted in red

3

General

Lipid peroxidation

This is a measure of antioxidant. The manuscript carried out quantitative analyses on total antioxidant capacities  

4

Methodology

Chemical characterization of the extract

All Authors confined that chemical characterization of crude extract add no value to the outcome of our finding. The values (nmr, HRMS, UV) will be too complex with no meaningful interpretation

5

General

All characterization should be mentioned in the manuscript with the original chromatograms

All chromatograms are obtainable as contained in PhD Thesis (Reference 9) on the bibliography section

Dr. Olugbenga Kayode POPOOLA

Correspondence Author

Reviewer 2 Report

Overall, a quality article addressing an innovative and interesting topic.

Nevertheless, please improve the following points before publication:

1/ keywords must not coincide with the title

2/ in the introduction should be articulated:

- the research problem

- aim

- research hypothesis

3/ in the Chemicals and reagents section - please list ALL reagents used during the research, along with the source of origin;

for reagents used in in vitro tests (enzymes, substrates), please provide distributor catalog numbers

4/ for ALL equipment and devices, please use the notation: name (mode, manufacturer, city, country)

5/ for percent concentrations, please add information m/v, v/v, etc.

6/ if a standard curve was used in the determination, please provide the range of analyte concentrations, the equation of the curve, its fit

7/ methodology:

- section 2.4.5 - how was the phosphate buffer prepared? what was it made of? In what was the substrate dissolved? What concentrations of cojic acid were used and in what was it dissolved?

- Section 2.4.6 - how was the tris-HCl buffer prepared? what was it prepared from? What type of elastase was used (e.g., I, III, etc.?) and what was it dissolved in? What concentration of oleanolic acid was used and what was it dissolved in?

8/ Table 3 - for results expressed as percentage of inhibition, there is no value greater than 100, writing >100 is incorrect and unacceptable! a lower sample concentration should have been used; please correct this; moreover, for % inhibition, the specific sample concentration should be given, which is crucial when we do not use IC50 as the unit

Minor editing of English language required

Author Response

AUTHORS REPLY TO THE REVIEW REPORT (REVIEWER 2)

Article No. Plants-2491004

Article type: Research

No.

Part

Comments

Author Response

1

Abstract

Keywords must not coincide with the title

Adjusted with inclusion of phytochemicals, while methoxylated and kaurane were expunged ……. all highlighted in red

2

Introduction

Introduction to be articulated in the area of:

The research problem

Aim

Research hypothesis

Adjusted with inclusion of my PhD Thesis where the research was carried out as Reference No. 9. This was also linked with the hypothesis …….all highlighted in red

3

Materials and methods

List all reagents used during the research, along with the source of origin

All reagents used has been listed with source of origin ……..all highlighted in red

4

Materials and methods

List all equipment and devices with notations: name (mode, manufacturer, city, country)

All equipment used to carry out this research has been included with necessary information

5

Materials and methods

For percent concentration, add m/v

All percent concentration adjusted and highlighted in red

6

Results

If a standard curve was used in the determination, please provide the range of analyte concentration, the equation of the curve its fit

1.      ORAC, FRAP, and TEAC values (Table 2) were determined using their respective templates (this is automated generated values from the computer after inputting all required data obtained from the analyses).

2.      Fe2+-induced lipid peroxidation, anti-tyrosinase and anti-elastatse  assay calculations, expressed as percentage inhibitions ± SD, were performed using MS Excel 2013, while the final values expressed as IC50, were determined using GraphPad prism 5.0 (Tables 2 & 3)

7

Methodology

Section 2.4.5: How was the phosphate? What was it made of? In what was the substrate dissolved? What concentrations of kojic acid were used and in what was it dissolved

1.      Authors confined that detailed preparation of buffers were captured in the references quoted: for example References 16-18 fully described LPO, TYR and ELA methodologies

2.      Kojic acid (positive control) concentrations were prepared with same concentrations of test samples. ……..this has been included in parenthesis indicated with red colour

3.      Kojic acid was dissolved in DMSO, same solvent used for test samples to a concentration of 1 mg/mL, further diluted with buffer to the effective concentrations as reported. Effected with red colour

8

Methodology

Section 2.4.6: How was Tris-HCl prepared? What was it prepared from, what type of elastase was used and what was it dissolved in, what concentration of oleanolic acid was used and what was it dissolved in

1.      Authors confined that detailed preparation of buffers were captured in the references quoted: for example References 16-18 fully described LPO, TYR and ELA methodologies

2.      Elastatse from porcine pancrease ……..details has been included with Sigma-Aldrich CAS No.

3.      Both positive control and test samples were first dissolved in methanol 1 mg/mL (m/v), further diluted to 100.00, 50.00, 25.00, 12.50, 6.25, and 3.12 µg/mL concentrations in tris-HCl buffer as working solutions

9

Results

Table 3

As reported in sample analyses section ditto to No. 6 above. All the Antioxidant (ORAC, FRAP and TEAC) results were reported as computer automated using their respective templates, while Inhibition of LPO, TYR and ELA were reported as IC50 using GraphPad 5.0

Dr. Olugbenga Kayode POPOOLA

Correspondence Author

Reviewer 3 Report

1. The plant material was collected in October 2012 from Jonkershoek - I don't understand why the plant was collected 11 years ago

2. Identification of seven known compounds I don't think it's relevant.

3. It should be justified why a methanolic extract was chosen

4.The conclusion that it can be used as a cosmetic product also requires an in vivo study

5.The bibliography does not include sufficient studies on cosmetic products obtained from plants. Discussions related to this aspect are missing

Author Response

AUTHORS REPLY TO THE REVIEW REPORT (REVIEWER 3)

Article No. Plants-2491004

Article type: Research

No.

Part

Comments

Author Response

1

Materials and methods

The plant material was collected in October, 2012 from Jonkershoek

This manuscript was part of my PhD findings between 2101-2015. The Editor also suggested inclusion of the PhD Thesis which have been referenced in the bibliography as Reference No. 9 and cited in the introductory section ……..all highlighted in red

2

General section

Identification of seven known compounds

All Authors confined that the word “identification” was ok. We carried out spectroscopic (NMR, UV, HRMS) analyses, followed by data interpretation and correlation with established data ……… we think identification of known compounds is proper

3

Materials and methods

It should be justified why a methanolic extract was chosen

1.   We employed total extraction with polar solvent of methanol followed by fractionation with chromatographic techniques

2.   We believed that methanol can extract our desired phytochemicals (polyphenolic compounds) which are good antioxidants

3.   We are also interested in comparative analyses with other South African Helichrysum species using same methods of analyses with our previous publications

4

Conclusion

The conclusion that it can be used as a cosmetic product also requires an in vivo study

This has been included in concluding part and highlighted in red

5

References

The bibliography does not include sufficient studies on cosmetic products obtainable from plants. Discussion related to this aspect are missing

The research was designed to identify possibility of sourcing for natural raw materials for cosmetic products formulation but limited to degenerative action of skin enzymes and over accumulation of free radicals. Therefore, references Nos. 4,5, 18, 25 (newly included), 40 & 41 pointed to information about sourcing for natural skin anti-aging raw materials from plants

Dr. Olugbenga Kayode POPOOLA

Correspondence Author

Round 2

Reviewer 1 Report

This revised article is not appropriate for publication, since the authors didn't answer well in my urgent and vital comments,

1. Chemical characterization of the extract

2. All characterization should be mentioned in the manuscript with the original chromatograms

The originality and novelty in this work are missing.

Minor errors

Author Response

AUTHORS REPLY TO THE REVIEW REPORT (REVIEWER 1 ROUND 2)

Article No. Plants-2491004

Article type: Research

No.

Part

Comments

Author Response

1

Materials and methods

Authors should conduct chemical characterization of the extract

The major chemical characterization carried out was Nuclear Magnetic Resonance (nmr). The spectroscopic determination of structures is not applicable to extract but only isolated compounds. Spectral obtain form nmr of extract are not interpretable due to multiple overlap of signals

5

Materials and methods

All characterization should be mentioned in the manuscript with the original chromatograms

All chromatograms (TLC and HPLC) are now included in section 2.3, Figures 1a-1i, while description of such figures were highlighted yellow in same section

Dr. Olugbenga Kayode POPOOLA

Correspondence Author

Reviewer 2 Report

Accept in present form

Author Response

AUTHORS REPLY TO THE REVIEW REPORT (REVIEWER 2 ROUND 2)

Article No. Plants-2491004

Article type: Research

Reviewer 2 round 2 comment: The Reviewer 2 has declared the manuscript accepted for publication

Dr. Olugbenga Kayode POPOOLA

Correspondence Author

Round 3

Reviewer 1 Report

Accept

Minor errors